# Breaking Semantic Artifacts for Generalized AI-generated Image Detection

**Chende Zheng**[1]  **Chenhao Lin**[1*]  **Zhengyu Zhao**[1]  **Hang Wang**[2]
**Xu Guo**[3]  **Shuai Liu**[3*]  **Chao Shen**[1]
[1]School of Cyber Science and Engineering, Xi'an Jiaotong University
[2]School of Automation Science and Engineering, Xi'an Jiaotong University
[3]School of Software Engineering, Xi'an Jiaotong University
zhengchende@stu.xjtu.edu.cn
{linchenhao, zhengyu.zhao, cshangwang}@xjtu.edu.cn
xuguo@stu.xjtu.edu.cn
{sh_liu, chaoshen}@mail.xjtu.edu.cn

## Abstract

With the continuous evolution of AI-generated images, the generalized detection of them has become a crucial aspect of AI security. Existing detectors have focused on cross-generator generalization, while it remains unexplored whether these detectors can generalize across different image scenes, e.g., images from different datasets with different semantics. In this paper, we reveal that existing detectors suffer from substantial accuracy drops in such cross-scene generalization. In particular, we attribute their failures to "semantic artifacts" in both real and generated images, to which detectors may overfit. To break such "semantic artifacts", we propose a simple yet effective approach based on conducting an image patch shuffle and then training an end-to-end patch-based classifier. We conduct a comprehensive open-world evaluation on 31 test sets, covering 7 Generative Adversarial Networks, 18 (variants of) Diffusion Models, and another 6 CNN-based generative models. The results demonstrate that our approach outperforms previous approaches by 2.08% (absolute) on average regarding cross-scene detection accuracy. We also notice the superiority of our approach in open-world generalization, with an average accuracy improvement of 10.59% (absolute) across all test sets. Our code is available at *https://github.com/Zig-HS/FakeImageDetection*.

## 1   Introduction

Recently, the rapid development of AI-generated image technology has led to the emergence of synthetic images on the internet, raising significant concerns about AI security. These images exhibit remarkable diversity due to the continuous introduction of new generative architectures, e.g., from Generative Adversarial Networks (GANs) [1], Variational Auto-Encoders (VAEs) [2], to Diffusion Models (DMs) [3]. For now, Internet users can easily generate a large number of exquisite images using text prompts. As the quality and variety of these synthetic images continue to advance, developing a universal detector for fake images becomes a crucial aspect of AI security.

To determine whether an image is synthetic, existing studies typically train a binary classifier to assess the authenticity of an unknown image during inference. Previous methods have primarily focused on cross-generator detection. For instance, Wang et al. [4] developed a ResNet50 trained on images from proGAN [5], and then tested it on different GAN variants. Intriguingly, their findings suggested

---

*Corresponding Authors

38th Conference on Neural Information Processing Systems (NeurIPS 2024).

that aggressive augmentation strategies could significantly enhance the classifier's generalization ability. However, some recent studies [6, 7] have shown that Wang's approach does not extend well to images synthesized by unseen DMs, even when DM-generated images are included in the training set. To address the problem of this cross-generator generalization, recent advancements have focused on refining detection algorithms [8], using large-scale pre-trained models [7, 9, 10], and augmenting or obtaining diverse datasets [11, 12]. In addition, Corvi et al. [13] demonstrated that images of one specific generative model contain unique artifacts that differ from those of other generators. These "generator artifacts" are particularly distinct between generators with different architectures, such as DMs and GANs, providing a promising avenue for improving cross-generator generalization.

Despite these efforts, it remains unexplored whether these detectors can generalize across different image scenes, e.g., images from different datasets with different semantics. Notably, Dogoulis et al. [14] have discussed a similar setting. However, their study focuses solely on different concept classes (e.g., objects) and does not consider a wider range of content. To fill this research gap, we propose a more comprehensive cross-scene problem and solution.

By analyzing the residual spectrum of images, we identify a significant challenge arising from the "semantic artifacts" in both real and generated images. Our findings reveal that images from different datasets contain unique artifacts, which can be inherited by generative models. During the training, existing detectors tend to overfit the specific artifacts of the training data, resulting in substantial accuracy drops in cross-scene generalization. For example, classifiers trained on images generated by Latent Diffusion trained on LAION struggle to generalize to images generated by the same diffusion method trained on FFHQ. Intriguingly, while "semantic artifacts" significantly impact the decrease in accuracy, the Average Precision of detectors remains high. This issue was also observed in [12], but comprehensive exploration is lacking.

To address this challenge, our work focuses on mitigating the impact of "semantic artifacts" for generalized AI-generated image detection. Specifically, we experimentally find that detectors based on deep networks tend to focus more on the global semantics of images, implying their classification may rely on specific scenes. To counter this, we propose a simple yet effective approach based on conducting an image patch shuffle and then training an end-to-end patch-based classifier. Our approach aims to extract artifacts of generators in a local patch while breaking the global "semantic artifacts". Meanwhile, by reducing receptive fields, our approach minimizes the excessive learning of "generator artifacts", benefiting cross-generator detection. To comprehensively evaluate the generalization ability of our approach, we conduct an open-world evaluation on 31 test sets, covering 7 Generative Adversarial Networks, 18 (variants of) Diffusion Models, and another 6 CNN-based generative models. Our experiments demonstrate the effectiveness of our approach across extensive evaluations. Our main contributions can be summarized as follows:

- We innovatively identify "semantic artifacts" in cross-scene AI-generated image detection, which leads to poor generalization performance of existing detectors.

- We propose a simple yet effective patch-based approach, aiming at breaking the "semantic artifacts" for generalization detection. By extracting local features, our detector is able to reduce the impact of global semantics in images.

- Extensive experiments validate the effectiveness of our approach with an improvement of 3.81% average accuracy (absolute) in cross-scene detection (6 variants of Latent Diffusion with different real datasets) and an improvement of 6.74% average accuracy (absolute) in open-world generalization (all 31 test sets).

## 2 Related Work

### 2.1 Generative Models

Generative models aim at learning to create new samples from a given dataset, with the fundamental goal of capturing the underlying probability distribution mapping. Prominent approaches in this field include Variational Auto-Encoders (VAEs) [2], Auto-regressive Models [15], Flows-based Models [16], Generative Adversarial Networks (GANs) [1], and Diffusion Models (DMs) [3]. Early methods like GANs have achieved impressive realism in image synthesis on specific categories, leading to the development of the proGAN/styleGAN series [17, 18].

More recently, DM-based synthesis methods, exemplified by Latent Diffusion [19], have rapidly advanced in text-to-image generation. Breakthroughs like Stable Diffusion, DALL·E 2 [20], and Imagen [21] are released in quick succession. These large-scale image synthesis models have significantly expanded the application scope of image synthesis, enriching the semantic content and stylistic features of generated images. The evolution makes the detection of AI-generated images increasingly complex and challenging.

## 2.2 Detecting Synthetic Images

Prior to the advent of advanced DMs, a significant body of forgery detection research focused on images generated by GANs. Early detection methods targeted local artifacts present in synthetic images, including distortions in facial tampered landmarks [22] or inconsistencies in head postures [23]. In [24], it was demonstrated that global artifacts in synthesized images differ markedly from common artifacts found in modern digital devices. Additionally, the up-sampling operations prevalent in most GAN architectures tend to produce distinct peaks in the spectral profiles of synthesized images [25, 26], offering another avenue for detecting fake images.

A noteworthy question that arises is whether these methods are also effective in detecting DM-generated images. Wang [4] suggests that simple classifiers when trained with aggressive data augmentation on proGAN images, can be adapted to other unseen models. Ojha [7] use a fixed pre-trained CLIP:ViT-L/14 to extract image features, while Sha [27] incorporate hint information and utilize multimodal inputs (text prompts and images) to enhance performance. Tan et al. [8] introduce the concept of neighboring pixel relationships as a means to capture and characterize the generalized structural artifacts stemming from up-sampling operations. In particular, recent studies [28, 29] have identified significant differences in the frequency spectra of images synthesized by GANs and DMs, impacting the generalization capability of existing detectors. Interestingly, a common issue is observed when detectors are extended to unknown data [12]. Specifically, these detectors will show a disparity where high Average Precision is accompanied by low accuracy. Our analysis of generalization detection of AI-generated images reveals a similar problem, showing performance imbalances when dealing with different semantics.

## 3 Methodology

### 3.1 Artifacts Analysis

To develop a universal detector for AI-generated images, it is crucial to thoroughly examine the distinctive artifacts of synthetic images generated by various generative models. Inspired by [4, 13, 7], we start our analysis by visualizing the frequency spectra of different image distributions. This exploration aims to uncover any unique or intriguing properties of the artifacts associated with different generators or scenes.

In line with the experimental design in [30, 29], we randomly select a set of images with a quantity of $I = 1000$ for each image source and utilize the denoising filter $D(x_i)$ from [31] to extract the noise residuals $r_i$ of an original image $x_i$:

$$r_i(m, n) = x_i(m, n) - D(x_i(m, n)) \quad i = 1, 2, ..., I \tag{1}$$

Then, we start from the Fourier transform $F$ of the noise residuals of the $M \times N$ image:

$$F_i(k, l) = \sum_{m=1}^{M} \sum_{n=1}^{N} r_i(m, n) e^{-j2\pi(\frac{k}{M}m + \frac{l}{N}n)} \tag{2}$$

where, $k$ and $l$ represent the frequency domain coordinates. To obtain the artifacts of the image source, we take the average power spectrum $S$ of all single images from it:

$$S_x(k, l) = \frac{1}{I} \sum_{i=1}^{I} |F_i(k, l)|^2 \tag{3}$$

We use images from our test sets (see Section 4.1) to conduct preliminary experiments for analysis of the artifacts in both synthetic and real images.

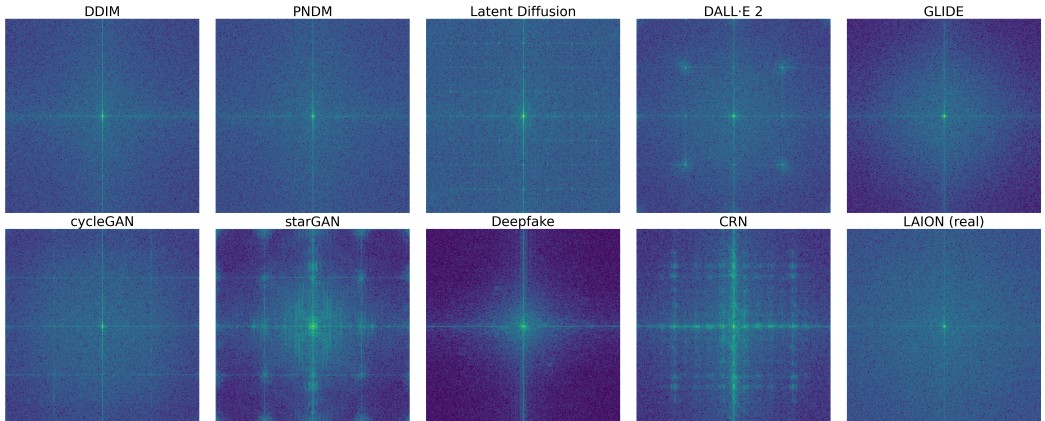

Figure 1: **Generator artifacts:** noise residuals power spectrum of images from 9 generative models and 1 real dataset. Top row: 5 Diffusion Models. Bottom row: 2 GANs, cycleGAN and starGAN, 2 CNN-based generators, Deepfake and CRN, and 1 real dataset, LAION.

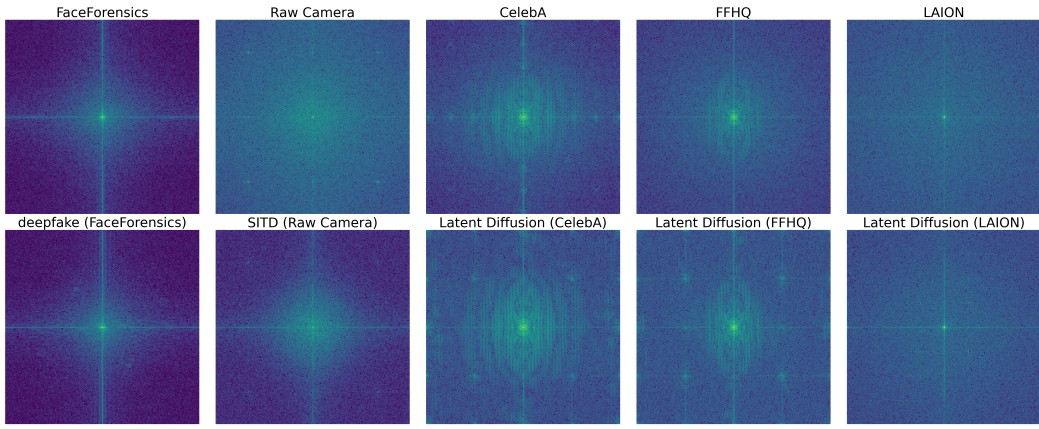

Figure 2: **Semantic artifacts:** noise residuals power spectrum of images from different scenes. Top row: 5 real datasets. Bottom row: 5 generative models in corresponding scenes, deepfake, SITD, and 3 variants of Latent Diffusion on CelebA, FFHQ, and LAION.

**Generator artifacts.**   Figure 1 shows the noise residuals power spectrum of generated images. For GANs, we have observed periodic artifacts in the spectrum, identified as checkerboard artifacts produced by up-sampling [25]. Similar artifacts are also observed in some DMs, such as DALL·E 2 and Latent Diffusion. These DMs employ an auto-encoder architecture to compress images into latent space features, resulting in periodic artifacts akin to those found in GANs. While other CNN-based generators, e.g., Deepfake or CRN, exhibit unique artifacts that are significantly different from GANs or DMs. Interestingly, although cycleGAN and starGAN are both based on GAN architecture, they still show some differences in the spectrum. Overall, images from different generators exhibit different artifacts and generated images with unique artifacts might lead detectors to overfit during training, thereby failing to recognize distinct artifacts of cross-generator data. We term these artifacts as "generator artifacts."

**Semantic artifacts.**   Figure 2 shows the noise residuals power spectrum of real and synthetic images from 5 distinct scenes (FaceForensics, Raw Camera, CelebA, FFHQ, LAION). To our surprise, artifacts are also observed in real images. For example, compared to LAION, CelebA and FFHQ exhibit artifacts around the center of the spectrum while Raw Camera displays artifacts similar to the periodic artifacts in GANs. Meanwhile, our findings suggest that these artifacts can be inherited by generative models (see Deepfake trained on FaceForensics, SITD trained on Raw Camera, and Latent Diffusion trained on CelebA/FFHQ/LAION). These artifacts might originate from the semantics of the images or the preprocessing methods of the datasets. For example, CelebA and FFHQ share

| Training | Metrics | Test | |
|---|---|---|---|
| | | Bedroom | Church |
| Bedroom | Acc. (Real) | 99.80 | 100.0 |
| | Acc. (Fake) | 100.0 | 0.00 |
| | AP | 100.0 | 99.90 |
| Church | Acc. (Real) | 100.0 | 100.0 |
| | Acc. (Fake) | 0.00 | 100.0 |
| | AP | 98.20 | 100.0 |

Table 1: Cross-scene detection experiments on ResNet-50 models from ForenSynths [4]. We use 2 sets of training images on different scenes, Bedroom and Church, to retrain the detectors. Detection accuracy (Acc.) (at a threshold of 50%) and Average Precision (AP) are reported.

| Training | Acc.@ | Test w/o patch shuffle | | Test w/ patch shuffle | |
|---|---|---|---|---|---|
| | | Bedroom | Church | Bedroom | Church |
| Bedroom | 50% | 99.99 | 50.00 | 100.0 | 51.18 |
| | 10% | 99.99 | 50.00 | 100.0 | 56.11 |
| | 1% | 99.96 | 50.07 | 99.94 | 69.24 |
| | 0.1% | 99.87 | 50.97 | 99.61 | 82.45 |
| | 0.01% | 99.27 | 62.03 | 97.94 | 91.59 |
| | 0.001% | 95.18 | 85.43 | 91.92 | 97.01 |
| Church | 50% | 50.00 | 100.0 | 94.15 | 100.0 |
| | 10% | 50.04 | 100.0 | 99.29 | 100.0 |
| | 1% | 52.67 | 99.96 | 98.71 | 99.96 |
| | 0.1% | 77.42 | 99.77 | 94.47 | 99.69 |
| | 0.01% | 95.32 | 99.29 | 84.64 | 96.99 |
| | 0.001% | 86.49 | 96.17 | 72.61 | 88.96 |

Table 2: Accuracy (at different thresholds) results of ResNet-50 models. We report the performance of the ResNet-50 models with and without the implementation of Patch Shuffle. Patch Shuffle is employed during both training and testing.

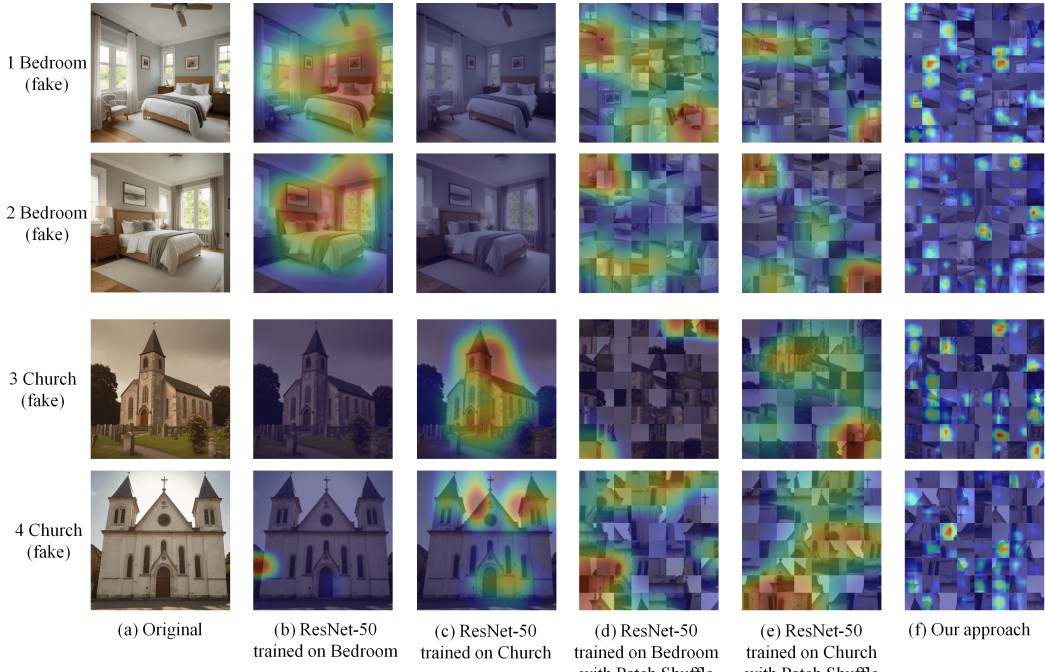

|        | (a) Original | (b) ResNet-50 trained on Bedroom | (c) ResNet-50 trained on Church | (d) ResNet-50 trained on Bedroom with Patch Shuffle | (e) ResNet-50 trained on Church with Patch Shuffle | (f) Our approach |

Figure 3: The visualization of CAM extracted from different detectors on Bedroom or Church images. Warmer color indicates a higher probability.

similar artifacts around the center of the spectra since both contain images of human faces. Notably, although FaceForensics is also a dataset of human faces, its significantly different preprocessing method (cropped from video and resized) results in unique artifacts. We term these artifacts as "semantic artifacts."

## 3.2 Analysis of the Impact of Semantic Artifacts

Many previous researches [10, 12, 8] have focused primarily on cross-generator detection of AI-generated images and have made good efforts on generalization across "generator artifacts". However, it remains unexplored whether these detectors can generalize on cross-scene detection, especially since we have found that "semantic artifacts" have many properties similar to "generator artifacts".

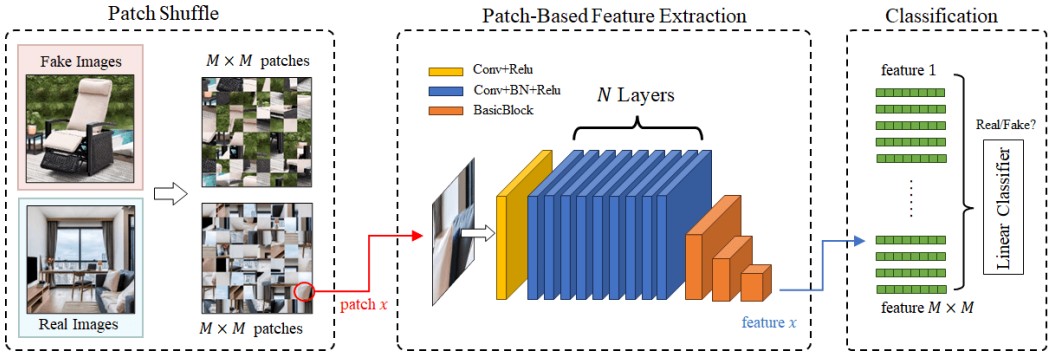

Figure 4: **Pipeline of our approach.** First, for pre-processing, we divide the input image into patches and shuffle these patches to obtain a randomized sequence. Then, we train a patch-based convolutional network for feature extraction. Finally, we flatten these features into a one-dimensional vector and then apply a linear classifier for classification.

To explore the impact of "semantic artifacts", we conduct several preliminary experiments using the existing paradigm [4]. This method is widely adopted in current studies [12, 10], where researchers train a ResNet-50 using images from only one generator to detect images from other generators.

Specifically, we use 2 sets of datasets for cross-scene tests. We randomly select 15000 images from LSUN bedroom and LSUN church, respectively, and generate 15000 images by SDXL-Turbo using prompts of "A photo of a bedroom" and "A photo of a church" respectively (10000 as training set and 5000 as test set). We train the detectors respectively on two training sets and evaluate them on cross-scene detection.

In Table 1, we report the accuracy (Acc.) and Average Precision (AP) of the 2 detectors on different scenes. As we can see, the detectors perform exceptionally well (almost 100%) on images with the same semantics as the training set. However, when tested on cross-scene images, the detectors suffer from significant drops in Acc. (near 0.00%) for generated images. Interestingly, the AP remains high, indicating that the detectors can still differentiate between real and fake images, although the distinction in scores between real and fake images is relatively minor. To further analyze the reasons for the low Acc., we report the Acc. results at different thresholds in Table 2. Typically, the threshold for determining whether an input image is real or fake is set to 50% in this task. However, in Table 2, we observe that the optimal threshold for AI-generated image detection is exceptionally low, ranging from 0.01% (from Church to Bedroom) to even 0.001% (from Bedroom to Church). This demonstrates that the average score of fake images with different semantics is extremely low and the score difference between real and fake images is minimal. Unfortunately, such a low threshold may lead to an increase in false alarms when handling various images. Furthermore, the optimal thresholds vary across different scenes, e.g., 0.01% from Church to Bedroom and 0.001% from Bedroom to Church. Therefore, simply adjusting the threshold does not resolve the issue of cross-scene generalization.

We attribute the poor performance of the existing paradigm to the overfitting of semantic artifacts during training. To address this, we simply apply Patch Shuffle as preprocessing to break the global semantics of images. The results are shown in Table 2. As observed, there is an improvement in Acc. at each threshold from Bedroom to Church, indicating a larger difference in scores between real and fake images. Moreover, from Church to Bedroom, the Acc. at the 50% threshold reaches 94.15%, highlighting the possibility of using Patch Shuffle to mitigate the impact of "semantic artifacts".

Figure 3 reports the visualization of CAM extracted from detectors on generated images of Bedroom or Church. Notably, detectors without Patch Shuffle struggle to focus on generated artifacts of images from different scenes. Due to the detector's limited ability to capture specific semantic content, there are almost no active regions in CAM of images from different scenes, e.g., (b)-3, (b)-4, (c)-1, (c)-2 in Figure 3. In contrast, detectors with Patch Shuffle demonstrate more active regions, unaffected by specific semantic content, e.g., (d)-3, (d)-4, (e)-1, (e)-2 in Figure 3.

### 3.3 Approach

The primary goal for cross-scene generalization is to break the impact of "semantic artifacts". In our preliminary experiments, simple Patch Shuffle has proven to be an effective method. However, overfitting still exists, e.g., from Bedroom to Church, the Acc. at 50% threshold of detectors with Patch Shuffle is 51.18% (see Table 2). We hypothesize that the deep residual structure of ResNet-50 promotes the extraction of overall semantic information. Although Patch Shuffle has been applied, the detectors still learn some semantic features between patches. A strong piece of evidence is that the visualization of CAM extracted from ResNet-50 with Patch Shuffle shows large, contiguous areas of activated regions, e.g., (d) and (e) in Figure 3, rather than discrete, point-like activated regions seen in our approach, e.g., (f) in Figure 3.

Based on this hypothesis, we propose a simple yet effective approach based on conducting an image Patch Shuffle and then training an end-to-end patch-based classifier. The whole pipeline of our approach is shown in Figure 4.

**Pre-processing.** First, given an input image $X$, we divide it into several patches of size $P \times P$:

$$\{p_i\} = Segment(X, P) \quad i = 1, 2, ..., M \times M \tag{4}$$

where $M \times M$ is the total number of patches obtained after segmentation. We further shuffle these patches to obtain a randomized sequence $\{p_j\}(j = 1, 2, ..., M \times M)$, which aims to break the global "semantics artifacts".

**Patch-based feature extraction.** We employ a patch-based convolutional network $\mathcal{C}$ for feature extraction, which is designed to accept only one single patch as input, aiming at extracting local features. For each patch $p_j$, we first apply a convolutional layer $C_0$ to extract an initial feature map $F_0$. Subsequently, a series of same-convolutional blocks $\{C_n\}$ are used for deep feature encoding.

$$F_0 = C_0(p_j) \tag{5}$$

$$F_n = C_n(F_{n-1}), \quad n = 1, 2, ..., N \tag{6}$$

where $N$ is a hyper-parameter representing the number of same-convolutional blocks. After encoding, the feature map is downsampled through 3 basicblocks.

**Linear classifier.** The features of each patch $\{F_N\}$ are reassembled into a feature sequence $\mathbf{L}$. We flatten the feature sequence into a one-dimensional vector $\mathbf{v}$, followed by a linear classifier $\mathcal{L}$:

$$\hat{y} = \mathcal{L}(\mathbf{v}) \tag{7}$$

We optimize the whole pipeline (including both the feature extractor and linear classifier) using binary cross-entropy loss:

$$\text{BCE}(y, \hat{y}) = -\frac{1}{N} \sum_{i=1}^{N} [y_i \log(\hat{y}_i) + (1 - y_i) \log(1 - \hat{y}_i)] \tag{8}$$

where $y$ represents the true labels, and $\hat{y}$ represents the predictions.

## 4 Experiments

### 4.1 Experimental Settings

**Training dataset.** To evaluate detectors' generalization capability, the standard practice is restricting the training data to images from only one generator and scene. We employ the generated images from DiffusionDB [32]. The training set consists of 48,000 images generated by Stable Diffusion v1.4 using prompts from Internet Users and 48,000 real images from LAION-5B [33]. To ensure a consistent basis for comparison, we also employ the training set of ForenSynths [4], in line with experimental design in [7, 10, 8], as an ablation experiment in Section A.3.1.

**Test datasets.** To evaluate the generalization of baselines and our approach, we conduct a comprehensive open-world evaluation on 31 test sets, covering 7 Generative Adversarial Networks, 18 (variants of) Diffusion Models, and another 6 CNN-based generative models.

- **Diffusion Models.** We collect DMs, including iDDPM [34], DDIM [35], PNDM [36], Guided-Diffusion (ADM) [37], RDM [38], Latent Diffusion (LDM) [19], Stable Diffusion v1.4 (SDv1) and v2.1 (SDv2), GLIDE [39], and DALL·E 2 [20]. We sample images from their variants of 6 real datasets, including LSUN-bedroom, LSUN-church [40], CelebA [41], FFHQ [17], ImageNet [42], and LAION-5B [33]. Details refer to Section A.7.

- **Generative Adversarial Networks.** The test sets include fake images generated by 7 GANs from ForenSynths [4], including proGAN [5], styleGAN, styleGAN2 [17], bigGAN [43], cycleGAN [44], starGAN [45], and gauGAN [46].

- **Other CNN-based generative models.** The test sets include fake images generated by 6 other CNN-based generative models from ForenSynths [4], including CRN [47], IMLE [48], SAN [49], SITD [50], deepfake [51], and WFIR [52].

**Detector baselines.** We perform comparisons of our approach with existing popular and state-of-the-art detectors on our open-world datasets, including Durall (CVPR 2020) [53], CNNDetection (CVPR 2020) [4], PatchFor (ECCV 2020) [6], F3Net (ECCV 2020) [54], Dogoulis (MAD'23) [14], DIRE (ICCV 2023) [55], Ojha (CVPR 2023) [7], LGrad (CVPR 2023) [10], NPR (CVPR 2024) [8]. We re-implement baselines [53, 4, 6, 54, 7, 10, 8] with the official codes using our training set, and adopt the official pre-trained weights of baselines [55].

**Implementation details.** Our approach is implemented using the PyTorch on NVIDIA A100 40GB Tensor Core GPU. During the training, we perform zero padding on the images to ensure the shorter edge is 256 pixels (resizing is also used as another pre-processing pipeline for ablation), and then randomly crop the images to 256x256. Each image will be horizontally or vertically flipped with a probability of 50% as data augmentation. The number of same-convolution blocks $N$ is set to 18 and the size of each patch $P$ is set to 32. The detector is trained

| Methods | Variants | Cross-Scene | | Open-World | |
|---|---|---|---|---|---|
| | | Avg. Acc. | mAP | Avg. Acc. | mAP |
| CNN (CVPR'20) | Blur+JEPG (0.1) | 53.66 | 63.40 | 56.23 | 65.27 |
| | Blur+JEPG (0.5) | 54.16 | 67.02 | 55.34 | 64.13 |
| PatchFor (ECCV'20) | ResNet18 | 66.13 | 74.67 | 57.68 | 63.76 |
| | Xception | 69.91 | 75.26 | 57.82 | 64.27 |
| F3Net (ECCV'20) | F3Net | 65.21 | 81.49 | 56.97 | 68.87 |
| | LFS | 57.53 | 79.36 | 55.07 | 76.66 |
| | Both | 56.82 | 85.39 | 53.93 | 74.54 |
| Durall (CVPR'20) | SVM | 64.23 | 59.80 | 55.87 | 53.76 |
| | LR | 68.28 | 64.81 | 59.15 | 53.74 |
| DIRE (ICCV'20) | CelebA-SDv2 | 63.19 | 69.93 | 57.05 | 63.81 |
| | ImageNet-ADM | 58.35 | 66.73 | 54.32 | 59.07 |
| | LSUN-ADM | 66.64 | 65.66 | 56.29 | 56.70 |
| Dogoulis (MAD'23) | Top 10k | 52.70 | 55.75 | 53.59 | 56.96 |
| | Top 24k | 51.91 | 57.76 | 53.25 | 60.35 |
| Ojha (CVPR'23) | CLIP:ViT-L/14+FC | 66.38 | 79.36 | 67.12 | 81.28 |
| LGrad (CVPR'23) | - | 62.91 | 80.99 | 57.69 | 76.91 |
| NPR (CVPR'24) | - | 90.44 | 94.84 | 75.38 | 88.76 |
| Ours | Resizing | **94.25** | **96.28** | 82.12 | 88.36 |
| | Zero padding | 92.52 | 95.58 | **85.97** | **90.00** |

Table 3: Results of cross-scene generalization and open-world generalization. For cross-scene generalization, we average the results on 6 variants of Latent Diffusion (LSUN-Bedroom, LSUN-Church, ImageNet, CelebA, FFHQ, LAION). For open-world generalization, we average the results on all 31 test sets (including 18 DMs, 7 GANs, and 6 CNN-based generators). **Bold** represents the best and underline represents the second best. More Detailed results are shown in Table 4 and Table 5.

using the Adam optimizer and early-stop strategy with an initial learning rate of $1e - 4$, a minimum learning rate of $1e - 6$, and a batch size of 64. We separate 5,000 images from the training set to serve as the validation set. We use the accuracy (Acc.) and the Average Precision (AP) as the evaluation metrics, with a particular emphasis on Acc. as our primary metric, because as discussed in Section 3.2, the AP metric may not fully reflect the problem of cross-scene generalization.

## 4.2 Experimental Results

**Cross-scene generalization.** We first report the cross-scene detection results in Table 3. We average the results on 6 variants of Latent Diffusion (LSUN-Bedroom, LSUN-Church, ImageNet, CelebA, FFHQ, LAION). The results demonstrate that most detectors suffer from performance drops on cross-scene images, even though the images are generated by models with the same structure. Particularly, the gap between AP and Acc. can be observed in several baselines, e.g., CNN (54.16% Acc. and 67.02% AP), Ojha (66.38% Acc. and 79.36% AP), F3Net (56.82% Acc. and 85.39% AP), LGrad (62.91% Acc. and 80.99% AP), which is consistent with our analysis in Section 3.2. As a comparison, the best baseline NPR and our approach show little gap between Acc. and AP. Considering that NPR uses Neighboring Pixel Relationships as features, which benefits breaking the global semantic artifacts, its high performance (90.44% Acc. and 94.84% AP) aligns with our hypothesis. Despite

| Methods | Variants | Bedroom | | | | Church | | | ImageNet | | CelebA | | FFHQ | LAION | | | | | | Average | |
|---|---|---|---|---|---|---|---|---|---|---|---|---|---|---|---|---|---|---|---|---|---|
| | | DDIM Acc. | iDDPM Acc. | PNDM Acc. | LDM Acc. | DDIM Acc. | PNDM Acc. | LDM Acc. | LDM Acc. | ADM Acc. | LDM Acc. | RDM Acc. | LDM Acc. | DALLE2 Acc. | GLIDE Acc. | LDM Acc. | SDv1 Acc. | SDv2 Acc. | SDv2-HR Acc. | Acc. | AP |
| CNN | Blur+JEPG(0.1) | 52.80 | 50.70 | 50.85 | 55.60 | 50.85 | 51.10 | 50.00 | 52.10 | 50.65 | 52.95 | 47.40 | 49.60 | 49.40 | 50.40 | 61.68 | 87.30 | 72.65 | 60.70 | 55.37 | 65.20 |
| | Blur+JEPG(0.5) | 50.75 | 52.00 | 50.20 | 52.75 | 50.75 | 50.35 | 50.35 | 52.60 | 49.95 | 57.40 | 48.05 | 50.90 | 50.00 | 50.15 | 60.93 | 84.40 | 75.50 | 66.40 | 55.75 | 66.91 |
| PatchFor | ResNet18 | 69.85 | 46.35 | 68.00 | 73.55 | 77.45 | 72.40 | 57.20 | 56.15 | 40.65 | 77.10 | 63.65 | 64.45 | 47.80 | 55.05 | 68.33 | 67.35 | 49.80 | 44.80 | 61.11 | 68.83 |
| | Xception | 50.10 | 51.35 | 50.15 | 51.95 | 51.30 | 50.15 | 50.75 | 66.80 | 49.95 | 95.90 | 50.00 | 54.80 | 50.70 | 60.00 | 99.25 | 99.35 | 68.40 | 67.95 | 62.16 | 69.14 |
| F3Net | F3Net | 50.15 | 50.00 | 50.00 | 50.15 | 50.15 | 50.00 | 50.00 | 60.35 | 42.55 | 89.05 | 52.85 | 76.85 | 50.45 | 49.95 | 64.88 | 86.20 | 76.70 | 84.20 | 60.25 | 77.31 |
| | LFS | 50.10 | 50.20 | 50.10 | 50.80 | 50.05 | 50.10 | 51.25 | 54.05 | 31.20 | 76.15 | 50.30 | 52.10 | 52.85 | 56.50 | 60.83 | 67.10 | 58.70 | 85.95 | 55.46 | 82.89 |
| | Both | 50.00 | 50.05 | 50.00 | 50.00 | 50.00 | 50.00 | 50.00 | 51.75 | 47.25 | 65.35 | 50.00 | 67.45 | 50.00 | 52.60 | 56.38 | 71.10 | 63.75 | 80.65 | 55.91 | 83.09 |
| Durall | SVM | 65.10 | 54.40 | 57.90 | 66.90 | 57.60 | 62.00 | 60.00 | 49.20 | 44.60 | 78.00 | 40.80 | 68.40 | 43.10 | 50.40 | 62.90 | 64.10 | 59.10 | 57.80 | 57.91 | 55.30 |
| | LR | 55.40 | 44.90 | 50.50 | 62.30 | 53.80 | 49.70 | 69.60 | 49.20 | 47.40 | 89.30 | 44.50 | 82.70 | 39.30 | 53.80 | 56.60 | 51.10 | 50.90 | 42.50 | 55.19 | 54.62 |
| DIRE | CelebA-SDv2 | 58.40 | 61.20 | 55.60 | 63.55 | 64.15 | 82.15 | 71.80 | 72.80 | 45.40 | 83.25 | 81.55 | 37.30 | 45.63 | 57.85 | 50.45 | 54.55 | 53.50 | 59.36 | 61.03 | 70.01 |
| | ImageNet-ADM | 50.00 | 51.70 | 49.70 | 47.90 | 51.85 | 51.90 | 49.25 | 73.95 | 42.70 | 81.80 | 81.45 | 50.05 | 54.22 | 63.95 | 47.14 | 43.85 | 45.95 | 52.77 | 55.01 | 60.90 |
| | LSUN-ADM | 50.00 | 49.95 | 50.40 | 50.20 | 50.60 | 51.00 | 50.45 | 96.65 | 46.30 | 99.90 | **100.0** | 49.65 | 52.01 | 53.25 | 53.01 | 53.30 | 53.85 | 75.67 | 60.34 | 61.92 |
| Dogoulis | Top 10k | 50.20 | 56.25 | 50.35 | 50.35 | 52.50 | 47.05 | 50.90 | 57.45 | **53.85** | 46.90 | 44.65 | 48.15 | 50.40 | 50.40 | 62.43 | 85.40 | 81.95 | 60.85 | 55.56 | 60.77 |
| | Top 24k | 50.35 | 51.30 | 49.90 | 50.10 | 51.30 | 49.45 | 49.95 | 53.15 | 52.20 | 47.25 | 46.15 | 49.55 | 51.40 | 51.30 | 61.48 | 89.10 | 82.20 | 59.90 | 55.34 | 63.49 |
| Ojha | CLIP:ViT-L/14+FC | 58.05 | 82.15 | 55.30 | 54.10 | 67.15 | 54.35 | 59.65 | 68.10 | 48.80 | 81.15 | 60.90 | 66.40 | 66.18 | 64.60 | 68.87 | 86.10 | 77.60 | 66.10 | 65.86 | 79.51 |
| LGrad | - | 56.90 | 59.90 | 54.20 | 51.15 | 51.60 | 54.25 | 50.35 | 58.10 | 36.65 | 96.90 | 64.25 | 57.95 | 53.75 | 58.10 | 63.03 | 77.65 | 68.60 | 68.71 | 60.11 | 85.78 |
| NPR | - | 52.80 | 56.90 | 54.60 | **99.75** | 59.20 | 54.60 | 83.65 | 92.30 | 44.15 | 99.90 | 97.55 | 68.45 | 77.28 | **90.15** | 98.60 | 96.20 | 94.85 | 89.30 | 78.35 | **94.08** |
| Ours | Resizing | 99.30 | **83.00** | **99.00** | 98.95 | **99.65** | **99.55** | **99.30** | 79.50 | 13.55 | **99.95** | 95.95 | **89.25** | 79.95 | 84.15 | 98.57 | 98.65 | 92.50 | 91.45 | 89.01 | 93.58 |
| | Zero padding | **99.40** | 80.40 | 98.65 | 93.45 | 96.70 | 98.20 | 82.70 | **97.40** | 26.70 | **99.95** | 93.15 | 81.95 | **88.83** | 89.90 | **99.69** | **99.70** | **97.60** | **98.90** | **90.18** | 93.64 |

Table 4: Cross-Diffusion generalization results. We evaluate the detectors on all 18 variants of Diffusion Models.

| Methods | Variants | proGAN | cycleGAN | bigGAN | styleGAN | styleGAN2 | gauGAN | starGAN | deepfake | SITD | SAN | CRN | IMLE | WFIR | Average | |
|---|---|---|---|---|---|---|---|---|---|---|---|---|---|---|---|---|
| | | Acc. | Acc. | Acc. | Acc. | Acc. | Acc. | Acc. | Acc. | Acc. | Acc. | Acc. | Acc. | Acc. | Acc. | AP |
| CNN | Blur+JEPG(0.1) | 51.00 | 50.00 | 49.60 | 49.80 | 51.16 | 65.95 | 50.20 | 62.25 | 51.11 | 56.62 | 74.45 | **85.35** | 48.80 | 57.41 | 65.38 |
| | Blur+JEPG(0.5) | 52.25 | 49.28 | 49.85 | 51.25 | 52.87 | 66.51 | 50.00 | 60.90 | 53.06 | 49.09 | 60.55 | 69.85 | 46.60 | 54.77 | 60.28 |
| PatchFor | ResNet18 | 54.25 | 52.73 | 53.35 | 52.65 | 59.26 | 62.92 | 58.70 | 56.95 | 46.67 | 50.23 | 43.50 | 46.90 | 49.95 | 52.93 | 56.73 |
| | Xception | 52.00 | 51.21 | 51.70 | 50.75 | 50.44 | 65.68 | 50.15 | 50.05 | 49.72 | 50.23 | 50.00 | 50.00 | 51.70 | 51.82 | 57.53 |
| F3Net | F3Net | 50.00 | 46.89 | 52.15 | 49.35 | 51.51 | 65.88 | 51.10 | 70.00 | 46.11 | 47.72 | 49.95 | 50.15 | 50.65 | 52.42 | 57.18 |
| | LFS | 64.25 | 50.45 | 50.75 | 50.20 | 53.06 | 65.06 | 50.15 | 50.00 | 50.00 | 39.95 | 57.25 | 68.65 | 59.15 | 54.53 | 68.04 |
| | Both | 50.25 | 50.00 | 49.85 | 49.95 | 47.66 | 65.51 | 50.00 | 55.40 | 41.94 | 47.95 | 50.10 | 51.05 | 55.90 | 51.20 | 62.70 |
| Durall | SVM | 21.20 | 58.20 | 55.40 | 57.70 | 73.80 | 49.40 | 94.70 | 50.00 | 19.50 | 16.20 | 45.20 | 52.20 | **96.30** | 53.06 | 51.63 |
| | LR | 80.50 | 56.70 | 54.40 | 57.20 | 74.50 | 48.90 | 79.00 | 55.90 | **82.00** | 76.40 | 57.30 | 65.00 | 52.40 | 64.63 | 52.51 |
| DIRE | CelebA-SDv2 | 48.44 | 53.80 | 49.80 | 52.55 | 54.65 | 19.65 | 50.95 | 50.05 | 49.43 | 65.50 | 64.15 | 63.50 | 47.50 | 51.54 | 55.24 |
| | ImageNet-ADM | 53.13 | 52.70 | 48.25 | 53.65 | 57.65 | 66.60 | 49.00 | 49.90 | 50.57 | 51.00 | 51.10 | 52.45 | 57.95 | 53.38 | 56.53 |
| | LSUN-ADM | 52.34 | 50.45 | 50.65 | 51.65 | 51.10 | 50.90 | 49.85 | 50.00 | 50.00 | 51.25 | 50.00 | 50.00 | 50.50 | 50.67 | 49.46 |
| Dogoulis | Top 10k | 51.00 | 49.55 | 49.50 | 48.05 | 48.93 | 65.99 | 50.00 | 50.00 | 43.06 | 50.91 | 48.40 | 56.05 | 49.75 | 50.86 | 51.68 |
| | Top 24k | 50.25 | 48.94 | 49.25 | 49.10 | 48.70 | 66.02 | 50.05 | 49.95 | 36.67 | 56.39 | 48.95 | 51.40 | 49.05 | 50.36 | 56.01 |
| Ojha | CLIP:ViT-L/14+FC | 91.25 | 74.90 | 79.05 | 84.75 | 71.25 | 73.05 | 72.30 | 62.05 | 49.72 | 64.38 | 50.50 | 53.15 | 68.90 | 68.87 | 83.74 |
| LGrad | - | 59.75 | 54.62 | 49.10 | 55.40 | 55.57 | 65.99 | 52.75 | 51.80 | 35.56 | 51.14 | 52.85 | 62.20 | 59.60 | 54.33 | 64.63 |
| NPR | - | **94.75** | **93.14** | 62.65 | 61.05 | 85.82 | 85.79 | **99.55** | 51.70 | 58.33 | 56.62 | 58.05 | 58.05 | 61.00 | 71.27 | 81.38 |
| Ours | Resizing | 86.50 | 84.77 | **89.85** | 90.25 | **88.85** | **91.46** | 96.50 | 66.80 | 10.28 | 60.00 | 47.90 | 58.55 | 71.85 | 72.58 | 81.12 |
| | Zero padding | 79.75 | 88.37 | 85.20 | **95.20** | 71.54 | 60.66 | **99.55** | **70.65** | 61.94 | **86.25** | **74.80** | 80.05 | 87.70 | **80.13** | **84.97** |

Table 5: Cross-GAN/CNN generalization results. We evaluate the detectors on all 7 Generative Adversarial Networks and 6 CNN-based generative models.

this, our approach still outperforms the SOTA generalization performance, showcasing higher Avg. Acc. and mAP metrics, which reach 92.52% and 95.58% (based on zero padding).

**Open-world generalization.** Table 3 also presents the Avg. Acc. and mAP metrics of detectors across the 31 test sets. The open-world evaluation results demonstrate an improvement in Avg. Acc. of our approach, outperforming Ojha and NPR by 18.85% and 10.59% respectively. This indicates the significance of breaking semantic artifacts and focusing on local features in generalized detection. The visualization result, e.g., the discrete, point-like activated regions in Figure 3 (f), further confirms the effectiveness of our approach that predominantly focuses on learning local features. Detailed results of our open-world experiments are presented in two groups.

- **Cross-Diffusion generalization.** Table 4 presents Acc. and mAP results across images from 18 (variants of) Diffusion Models. The results demonstrate the outstanding performance of our approach on cross-Diffusion evaluation. Our approach reaches an Avg. Acc. of 90.18%, outperforming the best baseline NPR (78.35%). Notably, most baselines show their poor performance on the unconditional generation scene, e.g., Bedroom and Church. We attribute it to a strong scene change, where there is a significant difference between the txt2img generation scene (training set) and the unconditional generation scene (test sets). In addition, the gap between Acc. and AP still exists, e.g., NPR (78.35% Avg. Acc. and 94.08% mAP).
- **Cross-GAN/CNN generalization.** Table 5 presents the Acc. and mAP metrics of detectors across images from 7 Generative Adversarial Networks and another 6 CNN-based generative models. As we see, most detectors suffer from performance degradation on cross-GAN/CNN Generalization, which is expected because these generators use a different architecture from

the Diffusion Model (training set). Nevertheless, our approach still shows advantages with SOTA accuracy while Ojha has SOTA mAP but the gap between Acc. and AP still exists.

**Effect of pre-processing pipeline.** So far, we have seen excellent generalizability in our patch-based method. We next explore in depth the influence of pre-processing pipelines on artifact altering and detection. Specifically, we adopt both image padding and resizing during pre-processing. Tables 4 and 5 demonstrate the superior results of zero padding compared to resizing, especially in cross-GAN/CNN generalization. In particular, the performance of our approach using padding has been largely boosted on SITD, SAN, CRN, IMEL, and WFIR compared to resizing. It can be attributed to their high image resolution, which introduces variations in artifacts and the loss of low-level features when resizing is applied. Meanwhile, Figure 5 in Section A.4 also supports this finding by showing that the frequency features of SITD (with image resolution of over 4,000×3,000) images change a lot after resizing. In summary, these results suggest that zero padding can preserve more low-level features in images, which benefits generalized detection.

**Effect of model depth and patch size.**
We next delve into the interplay between receptive field and generalization ability. Specifically, we adjust the number of same-convolution blocks ($N$), to vary the receptive field size and the quantity of parameters in the feature extractor (notably, resizing is used as preprocessing in the experiments because it does not affect the global receptive field). Table 6 reports the Avg. Acc. and mAP results across different $N$ values. The results demonstrate that both too-shallow ($N$=0) and too-deep ($N$=24) layers result in performance degradation due to underfitting and overfitting, respectively. Intriguingly, the best $N$ value for Cross-Diffusion generalization is 21 while the best for cross-GAN/CNN generalization is 6. Considering that images of the training set are generated by the Diffusion Model, we attribute the results to the fact that a shallow model might learn less

| Methods | Cross-Diffusion | | Cross-GAN/CNN | | Open-World | |
|---|---|---|---|---|---|---|
| | Avg. Acc. | mAP | Avg. Acc. | mAP | Avg. Acc. | mAP |
| P=64; N=0 | 81.53 | 87.48 | 63.55 | 66.42 | 73.99 | 78.65 |
| P=64; N=3 | 82.71 | 89.15 | 68.85 | 73.37 | 76.90 | 82.53 |
| P=64; N=6 | 86.34 | 90.96 | **74.27** | **80.40** | 81.28 | 86.53 |
| P=64; N=9 | 86.17 | 91.17 | 74.06 | **80.75** | 81.09 | **86.80** |
| P=64; N=12 | 87.09 | 91.85 | 73.37 | 79.52 | **81.34** | 86.68 |
| P=64; N=15 | 88.40 | 92.85 | 67.83 | 76.36 | 79.77 | 85.93 |
| P=64; N=18 | 88.60 | 92.43 | 67.96 | 72.93 | 79.95 | 84.25 |
| P=64; N=21 | **88.91** | **93.03** | 70.19 | 76.49 | 81.06 | 86.09 |
| P=64; N=24 | 87.10 | 91.65 | 67.70 | 72.28 | 78.97 | 83.53 |

Table 6: Ablation study results on model depth.

| Methods | Cross-Diffusion | | Cross-GAN/CNN | | Open-World | |
|---|---|---|---|---|---|---|
| | Avg. Acc. | mAP | Avg. Acc. | mAP | Avg. Acc. | mAP |
| P=64; N=18 | 88.60 | 92.43 | 67.96 | 72.93 | 79.95 | 84.25 |
| P=48; N=18 | 88.58 | 92.08 | 70.78 | 76.79 | 81.11 | 85.67 |
| P=32; N=18 | **89.01** | **93.58** | **72.58** | **81.12** | **82.12** | **88.36** |
| P=16; N=18 | 84.35 | 91.29 | 67.13 | 77.41 | 77.13 | 85.47 |

Table 7: Ablation study results on patch size.

from "generator artifacts" of training images and perform better on cross-generator detection. Conversely, this strategy will disrupt its fitting of in-domain artifacts, resulting in worse detection of in-domain datasets. Then, we conduct experiments to explore the effect of different patch sizes. The $P$ value represents the size of each patch, which determines the number of patches and the degree of breaking global semantic artifacts. Table 7 reports the Avg. Acc. and mAP metrics across different $P$ values. As expected, a larger $P$ value, e.g., $P$=48 or $P$=64, enlarges the receptive field, potentially exacerbating overfitting issues. Conversely, a too-small patch size, e.g., P=16, can lead to significant underfitting. This finding is encouraging, as it aligns with the effect of model depth and benefits a deeper investigation into the mechanisms of generalized detection.

## 5 Conclusion

In this paper, we introduce the concept of "semantic artifacts" in the generalization detection of AI-generated images. We analyze the existence and impact of "semantic artifacts" in cross-scene detection through extensive experiments. Our results demonstrate that existing detectors suffer significant drops in accuracy when applied to images with different semantics, which we attribute to overfitting during training. To address this issue, we propose a simple yet effective approach that utilizes Patch Shuffle and trains an end-to-end patch-based classifier to break the "semantic artifacts" in images. Extensive experiments on 31 test sets validate the effectiveness of our approach, demonstrating our contributions to the universal detection of AI-generated images.

## Acknowledgments

This research is supported by the National Key Research and Development Program of China (2023YFB3107401), the National Natural Science Foundation of China (T2341003, 62376210, 62161160337, 62132011, U21B2018, U20A20177, 62206217, 6240071050, 62406240), the Shaanxi Province Key Industry Innovation Program (2023-ZDLGY-38), the Natural Science Basic Research Plan in Shaanxi Province of China (2022JQ-631)

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

# A Appendix

## A.1 Limitations

This study has potential limitations. The effect estimates in the model are based on interventional and prospective observational studies. They are therefore subject to biases and confounding that may have influenced our model estimates. For instance, in Section 3.1, we qualitatively analyze the existence of semantic artifacts through visualization experiments and hypothesize that semantic artifacts might originate from the semantics of the images or the preprocessing methods of the datasets, where deeper theoretical proof is needed. In the future, we should develop quantitative metrics to systematically measure and evaluate semantic artifacts in image datasets.

## A.2 Broader Impact

This paper introduces a new method for enhancing the detection of AI-generated images. Socially, the research positively impacts security by better detecting synthetic media, thus mitigating risks like misinformation and fraud. However, it could also spur the development of more elusive generative models and raise privacy concerns if misapplied.

## A.3 Ablation Experiments

### A.3.1 Ablation studies on training data source.

To ensure a consistent basis for comparison, we also employ the training set of ForenSynths [4], in line with baselines [7, 10, 8]. The training set consists of 20 distinct categories, each comprising 18,000 synthetic images generated using ProGAN, alongside an equal number of real images sourced from the LSUN dataset.

Table 8 reports the Avg. Acc. and mAP results of detectors on our test sets. As expected, most detectors show better performance on cross-GAN/CNN generalization, due to the training set of proGAN. Despite this, our approach shows the best Avg. Acc. on cross-Diffusion, cross-GAN/CNN, and Open-World evaluation, which indicates the effectiveness of breaking semantic artifacts even on different datasets and the SOTA generalization ability of our approach.

### A.3.2 Ablation studies on training set size

Table 9 reports the Avg. Acc. and mAP results of our approach with different numbers of accessible training images. As expected, the increase in training set size will result in the improvement of

| Methods | Variants | Cross-Diffusion | | Cross-GAN/CNN | | Open-World | |
| --- | --- | --- | --- | --- | --- | --- | --- |
| | | Avg. Acc. | mAP | Avg. Acc. | mAP | Avg. Acc. | mAP |
| CNN | Blur+JEPG(0.1) | 54.28 | 69.03 | 81.07 | **93.31** | 65.51 | 79.21 |
| | Blur+JEPG(0.5) | 50.89 | 67.40 | 75.97 | 91.44 | 61.40 | 77.48 |
| PatchFor | ResNet18 | 66.57 | 80.02 | 74.29 | 85.71 | 69.81 | 82.41 |
| | Xception | 71.63 | 83.50 | 75.70 | 86.34 | 73.34 | 84.69 |
| F3Net | F3Net | 53.82 | 61.64 | 76.23 | 89.67 | 63.21 | 73.40 |
| | LFS | 70.72 | 84.98 | 74.84 | 80.89 | 72.37 | 83.27 |
| | Both | 67.89 | 86.15 | 75.58 | 82.94 | 71.11 | 84.80 |
| Durall | SVM | 52.34 | 53.82 | 52.38 | 50.07 | 52.36 | 52.25 |
| | LR | 50.69 | 52.18 | 58.95 | 49.01 | 54.15 | 50.85 |
| DIRE | *CelebA-SDv2* | *61.03* | *70.01* | *51.54* | *55.24* | *57.05* | *63.81* |
| | *ImageNet-ADM* | *55.01* | *60.90* | *53.38* | *56.53* | *54.32* | *59.07* |
| | *LSUN-ADM* | *60.34* | *61.92* | *50.67* | *49.46* | *56.29* | *56.70* |
| Ojha | CLIP:ViT-L/14+FC | 70.02 | 83.24 | 80.40 | 86.98 | 74.37 | 84.81 |
| NPR | - | 74.36 | **89.22** | 73.66 | 78.35 | 74.07 | 84.66 |
| Ours | P=32; N=18 | **81.80** | 85.58 | **81.80** | 85.07 | **81.80** | **85.37** |

Table 8: Open-world generalization results of detectors train on the dataset from ForenSynths [4]. *Italic* means that we adopt the pre-trained weights of DIRE, which are trained on diffusion-generated images, while other detectors are trained on proGAN-generated images.

| Methods | Cross-Diffusion | | Cross-GAN/CNN | | Open-World | |
|---|---|---|---|---|---|---|
| | Avg. Acc. | mAP | Avg. Acc. | mAP | Avg. Acc. | mAP |
| Training images = 10k | 80.00 | 88.26 | 67.24 | 73.54 | 74.65 | 82.09 |
| Training images = 20k | 81.20 | 90.34 | 69.36 | 79.42 | 76.24 | 85.76 |
| Training images = 40k | 88.84 | 93.45 | 70.60 | 80.00 | 81.19 | 87.81 |
| Training images = 80k | **88.93** | **93.87** | **71.67** | **82.06** | **81.69** | **88.92** |

Table 9: Ablation study results on training set size.

| Loss | Feature | Classifier | Open-World Avg. Acc. |
|---|---|---|---|
| BCE Loss | Patch-Based | Linear | **82.12** |
| MSE + BCE Loss | Patch-Based | Linear | 78.18 |
| BCE Loss | Fused | Linear | 69.51 |
| BCE Loss | Patch-Based | Self-Attention + Linear | 79.95 |

Table 10: Ablation study results on other components.

generalization. However, as the size of the training set reaches a certain scale, e.g., training images = 40k, such improvements gradually diminish.

### A.3.3 Other ablation studies

We consider that during the denoising process of Diffusion Models, the generated images might become overly "clean", lacking the environmental noise present in real-world scenes. To address this, we employ a loss for image denoising task on our patch-based feature extraction network instead of using Binary CrossEntropy (BCE) Loss. In the first stage, each patch is added with Gaussian noise of a certain intensity $y = x + v$ before being input into our feature extraction network. Then we use a Conv layer after the same-convolutional blocks to obtain the noise residual $R(y)$, thereby restoring the unnoticed patch $x' = y - R(y)$. We optimize the feature extractor by comparing the restored patch with the original patch by Mean Squared Error (MSE):

$$L_{\text{MSE}} = \frac{1}{2N} \sum_{i=1}^{N} ||R(y_i) - (y_i - x_i)||_F^2 \tag{9}$$

where $N$ represents the number of noisy-clean training patch pairs. Then, we optimize the linear classifier by BCE loss.

Additionally, we compare several other ablations: 1) Adding self-attention to the shuffled patch sequence before our linear classifier. 2) Using a pre-trained ResNet18 to extract global image features and fuse them with patch-based features for classification.

Our experimental results are shown in Table 10, where we notice that: **A.** When optimizing with MSE loss, we observe a decline in detection performance, which we attribute to two main reasons: 1) The added noise to the patches does not effectively replicate environmental noise. 2) This approach is not well-suited for images from GANs or other CNN-based generative models. **B.** The use of Self-Attention leads to reduced effectiveness, which might be due to the increase in model depth, inadvertently causing overfitting to artifacts. **C.** Using ResNet-18 for global feature extraction results in accuracy drops because this will introduce global semantic artifacts.

### A.4 Visualization of the effect of pre-processing pipelines

In this section, to better understand the impact of Patch Shuffled on artifacts, we analyze the noise residuals power spectra of shuffled images as well as image patches. As shown in Figure 5, these visualizations support our original hypothesis. Specifically, based on the visualization of shuffled images, most artifacts are removed during the patch shuffling. For instance, the distinct artifacts between CelebA and LAION or between LDM-CelebA and LDM-LAION are significantly reduced. In addition, the visualizations of image patches reveal an intriguing finding that low-frequency features are weakened but high-frequency features (corresponding to artifacts) are enhanced.

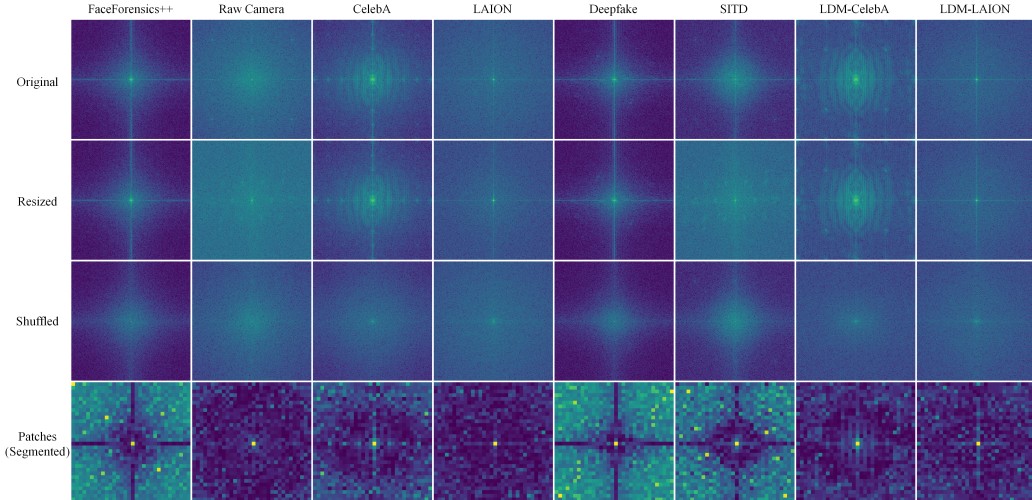

Figure 5: Additional noise residuals power spectrum of images from 4 real datasets (Faceforensics++, Raw Camera, CelebA, LAION) and 4 generative models (Deepfake, SITD, LDM-CelebA, LDM-LAION). As suggested by the reviewer, we consider 3 preprocessing operations from rows 2 to 4.

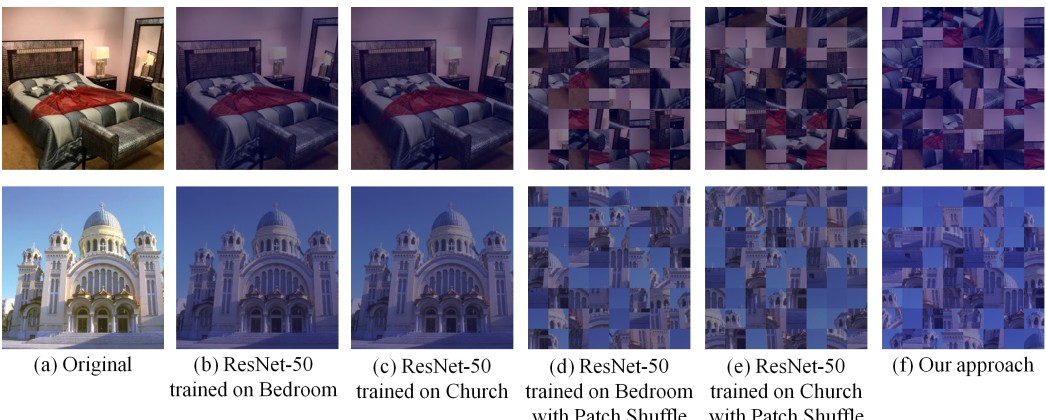

| (a) Original | (b) ResNet-50 trained on Bedroom | (c) ResNet-50 trained on Church | (d) ResNet-50 trained on Bedroom with Patch Shuffle | (e) ResNet-50 trained on Church with Patch Shuffle | (f) Our approach |

Figure 6: Additional CAM visualizations of real images for bedroom or church, where the label is "0". As expected, almost no region is activated, aligning with the high Acc. results of real images in Table 1.

## A.5 Visualizations of CAM extracted on real images

In this section, to better demonstrate the performance of the detector in cross-scene settings, the CAM visualizations of all the detectors on real images (i.e., with the label "0") are provided (see Figure 6). The visualizations show almost no activated regions, which aligns with the high Acc. results of real images in cross-scene settings (see Table 1).

## A.6 Analysis on Radial Spectrum

To refine the differences in spectral characteristics of various images, we analyze the radial power spectrum of image residual noise from different datasets. As usual, we obtain the Fourier transform $S_x(u, v)$ of the noise residuals from 1,000 images and normalize them (by dividing them by their maximum value). Then, we average the amplitude of frequency components with the same radial distance $r = \sqrt{u^2 + v^2}$ in the spectrum:

$$S(r) = \frac{1}{N(r)} \sum_{\sqrt{u^2+v^2}=r} |F_i(u, v)| \tag{10}$$

where $N(r)$ represents the number of frequency components at radius $r$, which goes from 0 to 1 in 180 discrete steps. $u$ and $v$ respectively represent the radial and lateral distances from the center of the spectrum. we categorize the radial spectra into low (10 60), mid (70 120), and high (130 180) frequency bands.

Figure 7 displays our experimental results, where the left column shows the low-frequency band results, the middle column for mid-frequency, and the right column for high-frequency. Rows 1 to 4 respectively exhibit the results for LAION, LSUN, ImageNet, CelebA, and their corresponding synthetic images, while Row 5 displays the results for LDM synthetic images trained on different datasets, and Row 6 shows results for different real dataset images.

In the results from Rows 1 to 4, we notice that most generative models exhibit characteristics close to real images in the low-frequency band. In the mid-frequency band, the differences become noticeable and are significantly amplified in the high-frequency band, indicating that the detection of synthetic images is traceable. On the other hand, we observe that real images always seem to exhibit smoother and more natural variations across all frequency bands. This is more evident in the results of the sixth row, where all classes of real images show the same trend and also have less jitter. In Row 5, we find that the frequency-level artifacts of generative models are concentrated at the highest components, and models using the same architecture tend to produce similar trends in variation. We observe similar trends in other results, such as DALLE2, SD-v1, and SD-v2 (Row 1); ADM (Row 3); and starGAN (Row 4). One explanation for this is that generative models with partly identical structures tend to produce similar artifact features. Nevertheless, when faced with a variety of generative methods, these artifacts will show significant differences, which further impact the generalizability of the classifier.

### A.7 Dataset Collection

We collect and sample a variety of DM-based generative models in chronological order of their release. 1) iDDPM [34], DDIM [35], and PNDM [36], which use the standard U-Net architecture and unconditional generation. 2) Guided-Diffusion (ADM) [37] and RDM [38], which use class-conditional generation. 3) Latent Diffusion (LDM) [19], Stable Diffusion v1.4 (SDv1) and v2.1 (SDv2), GLIDE [39], and DALL·E 2 [20] which use auto-encoders to edit images in latent space for txt2img generation. To explore detectors' cross-scene performance, we included variants of these DMs trained on 6 real image datasets: 1) LSUN-bedroom, LSUN-church [40], CelebA [41], and FFHQFFHQ [17] used for unconditional models; 2) ImageNet [42] used for class-conditional models, 3) LAION-5B [33] used for txt2img models. For class-conditional models, we uniformly sampled from each category, and for txt2img models, we used text from LAION as the prompts.

**iDDPM**[2]   We take the officially released iDDPM model pre-trained on LSUN bedroom and sample the synthetic images with *DDPM* sampler and *Linear* noise schedule. The number of diffusion steps is 1000. All the images are generated directly to $256 \times 256$.

**DDIM**   We use the implementation of DDIM pre-trained on LSUN bedroom and church from *https://github.com/luping-liu/PNDM* and sample the synthetic images with *DDIM* sampler and *Linear* noise schedule. The number of diffusion steps is 1000 with a speed of 50. All the images are generated directly to $256 \times 256$.

**PNDM**[3]   We take officially released PNDM models pre-trained on LSUN bedroom and church. Following the official code, we sample the synthetic images with *F-PNDM* sampler and *Linear* noise schedule. The number of diffusion steps is 1000 with a speed of 50. All the images are generated directly to $256 \times 256$.

**ADM**[4]   We take the officially released class-conditional Guided-Diffusion model pre-trained on ImageNet and sample the synthetic images with *DDIM* sampler and *Linear* noise schedule. The diffusion steps are 1000 and the respacing timesteps are 250. For each category (a total of 1000 categories) we generate images on average. All the images are generated directly to $256 \times 256$.

---

[2]https://github.com/openai/improved-diffusion
[3]https://github.com/luping-liu/PNDM
[4]https://github.com/openai/guided-diffusion

**RDM**[5]    We take the officially released Guided-Diffusion model pre-trained on CelebA and sample the synthetic images with *DDPM* sampler in the first stage and *Blurring* sampler in the second stage. The number of first steps is 256 and the second is 200. All the images are generated directly to $256 \times 256$.

---

[5]https://github.com/THUDM/RelayDiffusion

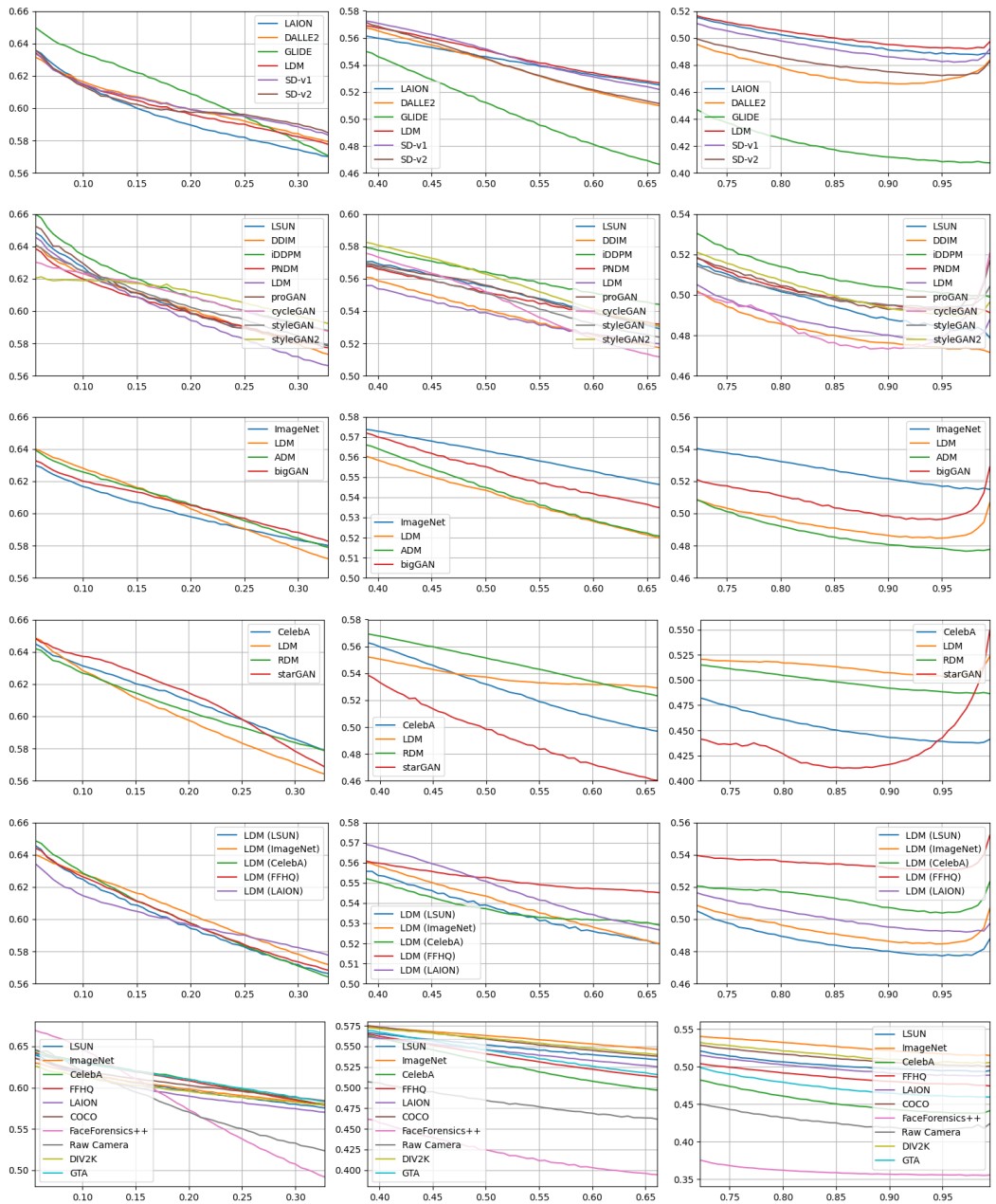

Figure 7: **Radial spectrum power density.** We calculate the radial spectra of different settings and divide them into 3 parts: low-frequency (left), mid-frequency (middle), and high-frequency (right). The x-axis represents the frequency components and the y-axis represents the power intensity of those components. Rows 1 to 4 show the results of images from LAION, LSUN, ImageNet, CelebA, and the generative models trained on the corresponding dataset. Row 5 shows the results of images from different LDM variants trained on LSUN, ImageNet, CelebA, FFHQ, and LAION. Row 6 shows the results of images from 10 different real datasets. Special artifacts of the generated model are more visible in the high-frequency range.

**LDM**[6]   We take officially released Latent Diffusion models pre-trained on CelebA-HQ, FFHQ, LSUN bedroom and church, ImageNet and LAION. Following the official code, we sample the synthetic images with *DDIM* sampler. The number of diffusion steps is 200. For each category of the ImageNet variant, we generate images on average and use prompts from LAION-High-Resolution and a CFG scale of 5.0 for the LAION variant. All the images are generated directly to $256 \times 256$.

**SD-v1**[7]   We take officially released Stable Diffusion v1.4 pre-trained on LAION and sample the synthetic images with *DDIM* sampler. The number of diffusion steps is 200. We use prompts from LAION-High-Resolution and a CFG scale of 5.0 for txt2img generation. All the images are generated directly to $256 \times 256$.

**SD-v2**[8]   We take officially released Stable Diffusion v2.1 models pre-trained on LAION and sample the synthetic images with *DDIM* sampler. The number of diffusion steps is 200. We use prompts from LAION-High-Resolution and a CFG scale of 5.0 for txt2img generation. All the images are generated directly to $256 \times 256$ and $768 \times 768$.

**GLIDE**[9]   We take the officially released GLIDE model and the corresponding upsampler. Following the official code, we sample the synthetic images with the default settings. The respacing timesteps of GLIDE are 250 and the steps of the upsampler are 100. We use prompts from LAION-High-Resolution for txt2img generation. All the images are generated to $64 \times 64$ and then upsampled to $256 \times 256$.

**DALL·E2**[10]   We take the implementation of DALL·E 2 pre-trained on LAION and sample the synthetic images with the default settings. We use prompts from LAION-High-Resolution for txt2img generation. All the images are generated directly to $256 \times 256$.

---

[6]https://github.com/CompVis/latent-diffusion

[7]https://github.com/CompVis/stable-diffusion

[8]https://github.com/Stability-AI/stablediffusion

[9]https://github.com/luping-liu/PNDM

[10]https://github.com/LAION-AI/dalle2-laion

