# OpenReview forum: "Breaking Semantic Artifacts for Generalized AI-generated Image Detection"
_NeurIPS.cc/2024/Conference — NeurIPS 2024 poster_

### Official Review · Reviewer_iKNb · 2024-06-27

**Soundness:** 2
**Presentation:** 2
**Contribution:** 2
**Rating:** 4
**Confidence:** 5

**Summary:**

This paper addresses the task of AI-generated image detection and specifically the challenge that arises by shifts in the semantics between training and test samples. It is demonstrated through experiments that different artifacts are produced by different generators, leading to performance drops in the cross-generator setting, and that the artifacts related to dataset semantics are inherited to the generated images leading to performance drops in the cross-scene setting. To this end a pipeline is proposed involving patch-based training instead of processing the whole-image or a patch-shuffled version of it. The performance of the trained detector is better than the baselines in the cross-scene setting and in the cross-generator setting (with ACC) but worse in the cross-generator setting (with AP).

**Strengths:**

S1-The problem of current detectors that overfit the dataset's semantics is important.
S2-The preliminary experimental analysis of artifacts is enlightening.
S3-The results support the effectiveness of the method.
S4-The experimental analysis (comparisons, ablations) is sufficient to understand the model's performance.

**Weaknesses:**

W1-The problem of performance drop in cross-scene settings has already been discussed in (Dogoulis et al. 2023).
W2-The method is extremely simple. A small CNN model processes the image patches (instead of the image), then aggregates the corresponding features and finally classifies the sample. Nothing methodologically novel. Also, its performance is very close and in some cases worse than the SotA NPR model. The initial findings seem very interesting, thus a more sophisticated approach would potentially perform even better in the cross-scene setting.
W3-Big performance gaps by this method between ACC and AP are still observed although it is claimed that this is what the method addresses. E.g., in Table 5 --> ACC 72.58 AP 81.12.
W4-Resizing the images to 256 reduces the synthetic artifacts, and it should be omitted. The fact that the training images are small makes the effect of resizing very small. I would like to see how much the results are changed if the authors omit this part of the data augmentation/pre-processing pipeline.

Dogoulis, P., Kordopatis-Zilos, G., Kompatsiaris, I., & Papadopoulos, S. (2023, June). Improving synthetically generated image detection in cross-concept settings. In Proceedings of the 2nd ACM International Workshop on Multimedia AI against Disinformation (pp. 28-35).

**Questions:**

What is the performance of the model if resizing is completely omitted?
How does the existing previous approach by Dogoulis et al. perform on your experimental design?

**Limitations:**

Sections 3.1 and 3.2 are not part of the methodology in my opinion, they should be placed under a separate motivation section as preliminary/insightful experiments. Section 3.3 although very small is still verbose and gives unnecessary details.
A similar setting has already been proposed by Dogoulis et al. 2023 which should be cited and the differences with the proposed definition/approach should be discussed.

---

> ### Author Rebuttal · Authors · 2024-08-05
>
> Thank you for taking the time to read our manuscript and for providing detailed comments! Below are our responses to the comments provided:
>
> >   Q1: Comparison to (Dogoulis et al. 2023), which has discussed the problem of performance drop in cross-scene settings.
>
> First, we acknowledge that the cross-scene generalization problem is well recognized, and our contribution is not to define the problem but to better solve it. Tables A and B in the PDF attached with the global response show that our method largely surpasses (Dogoulis et al. 2023) by **32.38\%** in Acc. and **29.65\%** in AP on open-world evaluation.
>
> See the following for the explanations of their poor performance and the experimental settings. The original setting in (Dogoulis et al. 2023) is less challenging because they only focus on generalization between different concept classes (e.g., objects) but do not involve more diverse contents, represented by the LAION, as in our work. Therefore, their assumption is essentially not sufficient for text-conditional generation. We notice that they originally select the top-10k images of 20k generated images as training sets, while we have 48k generated images. For a fair comparison, we consider two settings: A) top-10k images and B) top-24k images. We followed their official codes and selected ResNet-50 as the backbone. Gaussian blurring and JPEG compression are applied as data augmentation with a probability of 10\%.
>
> >   Q2: Missing discussion and comparison on the influence of resizing the images.
>
> See our response in Q3 and Q4 of Reviewer Zwma. In sum, we add ablation experiments and find that padding is a better choice than image resizing.
>
> >   Q3: Big performance gaps between Acc. and AP are still observed in Table 5.
>
> As seen in our response in Q3 and Q4 of Reviewer Zwma, the most considerable gaps on specific generators, e.g., SITD, SAN, CRN, IMLE, and WFIR can be solved by replacing the resizing with padding. For example, Table B shows that our method reaches the SOTA on cross-GAN/CNN (Acc. **80.13\%**, AP **84.97\%** vs. Acc. **72.58\%**, AP **81.38\%** of the second best method, NRP. The ACC-AP gap decreases from **8.8\%** to **4.84\%**.
>
> >   Q4: The method is extremely simple and nothing is methodologically novel. Close performance compared with the SOTA baseline.
>
> Based on our new results in Table B, our method substantially improves the previous SOTA, NPR, from ACC. 75.38\% to 85.97\% in the open-world evaluation.
>
> We welcome further improvements on top of our work. However, we believe the simplicity of our method does not diminish its novelty. Conceptually, our novelty lies in introducing the concept of "semantic artifacts" and demonstrating that breaking "semantic artifacts" is the key to cross-scene generalization. Based on this finding, we propose a novel design of a patch shuffle-based end-to-end detection framework.
>
> >   Q5: Sections 3.1 and 3.2 are not part of the methodology in my opinion, they should be placed under a separate motivation section as preliminary/insightful experiments. Section 3.3 although very small is still verbose and gives unnecessary details.
>
> We will further smooth the logic in Sections 3.1 and 3.2, and move unnecessary details to the Appendix.
>
> We agree that Sections 3.1 and 3.2 are not directly about the methodology, but we group them together with Section 3.3 to ensure a smooth transition from the motivation about "semantic artifact" to the method design.

---

> > ### Author Response · Authors · 2024-08-13
> >
> > Dear reviewer iKNb,
> >
> > We hope our rebuttal has addressed your concerns. We are looking forward to your reply and are more than willing to provide responses to any further inquiries you may have.
> >
> > Thank you very much.

---

### Official Review · Reviewer_4wrZ · 2024-07-12

**Soundness:** 3
**Presentation:** 3
**Contribution:** 3
**Rating:** 4
**Confidence:** 5

**Summary:**

This paper identifies a significant drop in accuracy for existing detectors when generalizing across different image scenes, attributing failures to "semantic artifacts." To address this, a new approach is proposed that involves image patch shuffling and training a patch-based classifier, enhancing generalization.

**Strengths:**

This paper is relatively well-motivated as AI-generated image detection is a crucial issue. I also find the evaluations thorough. The strengths are as follows:

The target issues of the paper are meaningful and worth exploring. The motivation is clear.
The paper is easy to follow.

**Weaknesses:**

1. The idea comparison of patchfor [6] and this paper is needed.
2. It is better to conduct an experiment using the images collected from fake news on the Internet.

**Questions:**

See weaknesses.

**Limitations:**

See limitations.

---

> ### Author Rebuttal · Authors · 2024-08-05
>
> Thank you for your comments! Below are our responses to the comments:
>
> >   Q1: Idea-level comparison to PatchFor [6].
>
> The key idea-level difference between PatchFor and our method is the use of patch shuffle. This is the core of our method, which ensures that the model completely removes the semantic information as a whole, thereby mitigating overfitting to semantic artifacts. Another technical difference is that we force our model to accept **only** patch inputs but PatchFor still accepts the full image, which contains rich semantic information.
>
> >   Q2: Experiments using the images collected from fake news on the Internet.
>
> We have already followed the common practice [4,7,8] for datasets and experimental settings. Notably, these datasets include data from the Internet, such as Deepfake (sourced from YouTube videos) and WFIR (sourced from websites). For the fake news, we believe it is clearly a valuable direction for future work but technically, it is infeasible because the ground truth of the fake news on the Internet is hard to get, making the model training and evaluation not possible.

---

> > ### Author Response · Authors · 2024-08-13
> >
> > Dear reviewer 4wrZ,
> >
> > We hope our rebuttal has addressed your concerns. We are looking forward to your reply and are more than willing to provide responses to any further inquiries you may have.
> >
> > Thank you very much.

---

### Official Review · Reviewer_Zwma · 2024-07-12

**Soundness:** 3
**Presentation:** 3
**Contribution:** 3
**Rating:** 6
**Confidence:** 5

**Summary:**

The paper looks into the problem of AI generated image detection. Given the onset of diffusion models they highlight how previous approaches which claim to generalize fail in setting with new datasets and models. The authors also motivate by visualizing frequency spectrum images. The paper then proposes a patch based feature extraction approach followed by a classifier which works on the image patches to classify whether the image is real or generated. Using such an approach is able to remove issues which some of the previous approaches faced and helps generalize better.

**Strengths:**

The following are the strengths of the paper:
1. The paper is very well written and easy to follow. The authors explain different components well.
2. The papers motivation is sound and the authors are able to highlight drawbacks of existing approaches well and also provide some proof backing their claims.
3. On highlighting the problems, the authors propose a new approach to handle the task and incorporate components in it which make it generalizable to new generators.
4. The authors have a good suite of generators that they evaluate in to show effectiveness of their approach and also showcase cross scene and cross model generalization.

**Weaknesses:**

The following are the weaknesses of the work:

1. The authors propose patch based learnings as a way to make the approach more generalizable. While they show frequency based visualizations for normal images, similar comparison with patch shuffled as well as image patches is missing.
2. In L178-180, it is not clear what artifacts authors are referring to. Also L192-193 again refers to the same Figure 3 but it is unclear why the visualization should be more focused on point regions. It could be that the detector feels the full region is the cause of its prediction.
3. In L242-243 the authors mention that they resize the images. Missing discussion whether this resizing alters the artifacts in the images. A visualization similar to frequency visualizations would help clarify this.
4. Missing analysis of why the proposed approaches has considerable gaps with existing approaches on specific generators. For example in Table 5 for many generators the proposed approach is far behind the best approach. Similar discussion for the quantiative results should be provided to give some intuitions.
5. Missing comparisons:
a) Missing comparison with Towards Discovery and Attribution of Open-world GAN Generated Images.
b) A previous work Towards Discovery and Attribution of Open-world GAN Generated Images also looked at fingerprints from generators. Missing comparison and discussion about it.

Small corrections:
L119 bot -> both
L119 synthetics -> synthetic

**Questions:**

I think the paper is well written and in general easy to follow and sound. At the same time I have mentioned a few clarification and comparisons in the weaknesses and would hope the authors answer some of them especially regarding more discussion about the presented quantitive numbers and missing comparisons with mentioned approaches. Some more analysis around frequency visualizations of patches and patch shuffled images would be good.

**Limitations:**

Yes.

---

> ### Author Rebuttal · Authors · 2024-08-05
>
> Thank you for the feedback and comments! We are glad to hear that you believe this is important work and that it meets the standard for publication. Below are our responses to the comments provided:
>
> >   Q1: Missing frequency visualizations on patch-shuffled images as well as image patches.
>
> Thank you for your suggestions. The frequency visualizations of shuffled images and image patches are provided in Figure A of the PDF attached with our global response.
>
> These visualizations support our original hypothesis. Specifically, based on the visualization of shuffled images, most artifacts are removed during the patch shuffling. For instance, the distinct artifacts between CelebA and LAION or between LDM-CelebA and LDM-LAION are significantly reduced. In addition, the visualizations of image patches reveal an intriguing finding that low-frequency features are weakened but high-frequency features (corresponding to artifacts) are enhanced.
>
> >   Q2: Not clear what artifacts are in L178-180 and why the visualization in Figure 3 should be more focused on point regions.
>
> We will make it clear that the "artifacts" in L178-180 refer to "generator artifacts", as defined in Section 3.1.
>
> For visualizations in Figure 3, we will clarify that the generator artifacts generally correspond to more universal, point regions, which represent a deeper receptive field, meaning a higher-level feature extraction. So it can be seen that our method successfully directs the model to focus on these regions rather than the concentrated, semantic regions.
>
>
> >   Q3: Missing discussion and visualization on the influence of image resizing.
>
> We acknowledge that the use of image resizing is not a particular design. In our additional experiments, we also test image padding and find it to be more effective. See Figure A and Tables A and B in the PDF attached with our global response.
>
> Tables A and B demonstrate the superior results of zero padding compared to resizing, particularly in cross-GAN/CNN generalization. In particular, the performance of our method has been largely boosted on SITD, SAN, CRN, IMEL, and WFIR. It can be attributed to their high image resolution, which introduces variations in artifacts and leads to the loss of low-level features when resizing is applied. Figure A supports this finding by showing that the frequency features of SITD (with image resolution of over 4,000×3,000) images change a lot after resizing.
>
> Note that on ganGAN, resizing is better than padding probably because their images have been resized during the original data collection.
>
> >   Q4: Missing analysis of why the proposed approaches have considerable gaps with existing approaches on specific generators.
>
> The considerable gaps (on SITD, SAN, CRN, IMEL, and WFIR) have been explained in the above response.
>
> >   Q5: Missing comparison and discussion about "Towards Discovery and Attribution of Open-world GAN Generated Images".
>
> The suggested paper and our work focus on different tasks. They aim at attribution and discovery of generated images, rather than our (generalized) detection. Therefore, a direct comparison is not possible. Technically, their method relies on out-of-distribution detection and clustering while ours directly destroys artifacts.
>
> >   Q6: Small corrections: L119 bot -> both L119 synthetics -> synthetic
>
> Thanks. We will fix them.

---

> > ### Comment · Reviewer_Zwma · 2024-08-12
> >
> > Thanks for the authors' response. I appreciate the authors' efforts in the rebuttal. Given the clarifications provided by the authors and the response and reviews to other reviewers, I will keep the positive initial rating I had provided.

---

> > > ### Author Response · Authors · 2024-08-12
> > >
> > > Thank you for acknowledging our rebuttal and supporting the acceptance of the paper. We will certainly incorporate your suggestions into the final version.

---

### Official Review · Reviewer_a5zv · 2024-07-13

**Soundness:** 2
**Presentation:** 2
**Contribution:** 3
**Rating:** 6
**Confidence:** 4

**Summary:**

The paper investigates the robustness and generalization of AI-generated image detectors, with a particular focus on the influence of semantic artifacts on detector performance. The authors highlight that semantic artifacts can significantly impact the effectiveness of AI detectors.
To explore this phenomenon, the authors employed a novel approach by training a classifier on image patches rather than whole images. By training a classifier on image patches, the authors conducted experiments on 31 different families of AI-generated images. This study provides a unique and intriguing perspective on how semantic artifacts can affect the performance and generalization of AI detectors.

**Strengths:**

Originality: The paper presents a unique angle by examining how semantic artifacts impact the generalization of AI image detectors. While this finding is not entirely unexpected, it represents a novel and insightful contribution to the field.

Quality: The paper successfully motivates the significance of semantic artifacts and their influence on generalization. However, the demonstration of this effect could be more thorough.

Clarity: The paper is well-written and includes visualizations that effectively aid in understanding the concepts and findings presented.

Significance: The paper demonstrates a notable increase in generalization performance, ranging from 3 to 6 points, across 31 test sets, including open-world scenarios. This improvement is significant and highlights the potential impact of addressing semantic artifacts in AI-generated image detection.

The paper evaluated on up to 8 baselines, while NPR outperforms on open-world, the results generally is good.

**Weaknesses:**

Missing Ablations: The impact of patch size on the performance of the classifier is not explored. Including an ablation study on different patch sizes would provide a deeper understanding of its influence on the results.

Missing Visualizations: Figure 3 lacks examples of real images, which is crucial for the reader to fully comprehend the differences and understand the context of the study. Including these examples would complete the visualization and enhance clarity.

Missing Data Details: The paper does not provide sufficient details on how real images were selected for the study. Additionally, there is a lack of discussion on various data design choices. Addressing these points would strengthen the paper by clarifying the methodology and justifying the design decisions.

Missing Citations: The paper should cite relevant works that discuss the use of patches to mitigate the influence of semantic features [1, 2].

[1] https://arxiv.org/abs/1905.13549
[2] https://openaccess.thecvf.com/content/CVPR2022/papers/Mao_Causal_Transportability_for_Visual_Recognition_CVPR_2022_paper.pdf

**Questions:**

Impact of Paired Data: What if paired data were used, where each real image is paired with an AI-generated image sharing the same semantic content? This pairing could be achieved by conditioning the generation process on the real image. By ensuring that both images share the same semantic features, we could eliminate semantic artifacts as a spurious factor. This approach might already solve the problem and lead to better generalization. The paper would benefit from discussing this potential strategy and its implications for improving the robustness of AI-generated image detectors.

---

> ### Author Rebuttal · Authors · 2024-08-05
>
> Thank you for taking the time to read our manuscript and for providing detailed comments! Below are our responses to your comments:
>
> >   Q1: Missing ablations of patch size on the performance.
>
> We have already reported the ablation results of patch size (and model depth) in Appendix A.1.1, and we will add conclusions in the main text based on them. The superiority of our method holds for all tested patch sizes and model depths.
>
> Specifically, a too-large patch size enlarges the receptive field, potentially exacerbating the overfitting issues, while a too-small patch size destroys certain low-level semantic features, potentially causing underfitting.
>
> >   Q2: Missing visualizations of real images in Figure 3.
>
> We add such visualizations to the attached PDF in our global response, as illustrated in Figure B, and we will include them in our updated manuscript.
>
> As expected, the CAM visualizations of all the detectors on real images (i.e., with the label "0") show almost no activated regions. This aligns with the high Acc. results of real images in cross-scene settings (see Table 1).
>
> >   Q3: Missing details about the data selection and design.
>
> We have already detailed the dataset collection in Appendix A.4, and we will summarize them in the main body of our updated manuscript.
>
> Specifically, for DM-generated images, our selected generative models span a long time range, from 2020 to now. They fall into 3 categories: unconditional, class-conditional, and text-conditional. For real images, the training data of each generative model are used to ensure the same distribution to their generated images.
>
> >   Q4: Missing citations of [1, 2], which discuss the use of patches to mitigate the influence of semantic features.
>
> Thank you for recommending these two papers, which can be used to better motivate our use of patches. We will cite them in our updated manuscript.
>
> >   Q5: Data pairing as a potential baseline.
>
> This is indeed a good point! A similar idea has been adopted by very recent methods, e.g., DIRE [54], which models the error between an input image and its reconstructed counterpart by a pre-trained diffusion model. We have validated that our method outperforms DIRE by **25.07\%** in Acc. and **24.55\%** in AP on open-world evaluation (see Table 3).
>
> We believe data pairing has the following limitations:
>
> +   It targets specific semantic objects, e.g., church, bedroom, human face, making it feasible only for unconditional or class-conditional generated images. However, for text-conditional generated images, which contain complicated semantics, paired data are hard to define and collect.
> +   Although paired data can remove the semantic artifacts, the model may still learn the domain-specific information. Then, when it comes to different domains during testing, the performance decreases. For example, a generated human face normally contains most artifacts in its domain-specific regions, e.g., the lineament and hair. This assumption is supported by our results in Table 4, where DIRE (trained on CelebA) degrades from **83.25\%** Acc. on CelebA-LDM to **50.45\%** Acc. on LAION-LDM.
>
> In contrast, our method is both tailored to specific semantics and indiscriminately destroys the semantic artifacts.

---

> > ### Comment · Reviewer_a5zv · 2024-08-07
> > **Thank you for the reply.**
> >
> > My questions are addressed and score updated.

---

> > > ### Author Response · Authors · 2024-08-11
> > >
> > > Thank you for your response and increasing the score. We will incorporate your feedback into the final version.

---

### Author Rebuttal · Authors · 2024-08-05

We thank all reviewers for their insightful assessment of our work and for providing useful feedback and actionable suggestions. They found that our research makes novel and insightful contributions (reviewer a5zv), with a clear and sound motivation (reviewers a5zv, Zwma, and 4wrZ), an effective/generalizable method (reviewers a5zv, Zwma, and iKNb), thorough/sufficient experiments (reviewers 4wrZ and iKNb), and good writing (all reviewers).

They mainly request more comparisons with recent research, discussion on the impact of preprocessing, and visualizations to further support our claims. To address them, we mainly provide:

● A comparison to the approach from "Dogoulis et al. Improving synthetically generated image detection in cross-concept settings" and an ablation of image resizing on the performance. (Tables A and B) (reviewer iKNb)

● Additional frequency visualizations on images from 4 real datasets and 4 generative models with different pre-processing pipelines. (Figure A) (reviewers Zwma and iKNb)

● CAM visualizations extracted from different detectors on real images of church or bedroom. (Figure B) (reviewer a5zv)

All these new results support our original claims. **In particular, thanks to the suggestion from reviewers, the performance of our method has been largely improved by replacing the resizing with padding.**

---

### Decision · Program_Chairs · 2024-09-25

**Decision:**

Accept (poster)

**Comment:**

This paper enhances the generalization of AI-generated image detectors by focusing on the challenge of cross-scene scenarios. Specifically, its key technique contribution is to train the classifier on image patches rather than whole images to mitigate the influence of semantic artifacts. Overall, the reviewers find this paper interesting to read and the provided empirical results are strong. But meanwhile, the reviewers raise several concerns, mainly regarding 1) the contribution/relationship w.r.t. some prior works should be discussed and clarified; 2) additional ablations (especially regarding image resizing) should be added; 3) some analyses need to be further discussed and clarified; 4) some critical visualizations are missing; and 5) the overall presentation can be further improved.

The provided rebuttal successfully alleviates most of these concerns, persuading two reviewers to rate this paper as 6 -- weak accept. The other two reviewers stand on the slightly negative side, rating this paper as 4 -- Borderline reject. By reading these two slightly negative reviews and the corresponding rebuttal, the AC believes most concerns are reasonably addressed and does not see any significant concerns remaining.

Given 1) the overall solidity of the submission and 2) the increasing demand for robust GenAI detectors, the AC would like to recommend accepting this submission, and believes this work will be of substantial interest to the NeurIPS community,